# A Minimax Optimal Algorithm for Crowdsourcing

**Thomas Bonald**
Telecom ParisTech
thomas.bonald@telecom-paristech.fr

**Richard Combes**
Centrale-Supelec / L2S
richard.combes@supelec.fr

## Abstract

We consider the problem of accurately estimating the reliability of workers based on noisy labels they provide, which is a fundamental question in crowdsourcing. We propose a novel lower bound on the minimax estimation error which applies to any estimation procedure. We further propose Triangular Estimation (TE), an algorithm for estimating the reliability of workers. TE has low complexity, may be implemented in a streaming setting when labels are provided by workers in real time, and does not rely on an iterative procedure. We prove that TE is minimax optimal and matches our lower bound. We conclude by assessing the performance of TE and other state-of-the-art algorithms on both synthetic and real-world data.

## 1 Introduction

The performance of many machine learning techniques, and in particular data classification, strongly depends on the quality of the labeled data used in the initial training phase. A common way to label new datasets is through crowdsourcing: many workers are asked to label data, typically texts or images, in exchange of some low payment. Of course, crowdsourcing is prone to errors due to the difficulty of some classification tasks, the low payment per task and the repetitive nature of the job. Some workers may even introduce errors on purpose. Thus it is essential to assign the same classification task to several workers and to learn the reliability of each worker through her past activity so as to minimize the overall error rate and to improve the quality of the labeled dataset.

Learning the reliability of each worker is a tough problem because the true label of each task, the so-called *ground truth*, is unknown; it is precisely the objective of crowdsourcing to guess the true label. Thus the reliability of each worker must be inferred from the comparison of her labels on some set of tasks with those of other workers on the same set of tasks.

In this paper, we consider binary labels and study the problem of estimating the workers reliability based on the answers they provide to tasks. We make two novel contributions to that problem:

(i) We derive a lower bound on the minimax estimation error which applies to any estimator of the workers reliability. In doing so we identify "hard" instances of the problem, and show that the minimax error depends on two factors: the reliability of the three most informative workers and the mean reliability of all workers.

(ii) We propose TE (Triangular Estimation), a novel algorithm for estimating the reliability of each worker based on the correlations between triplets of workers. We analyze the performance of TE and prove that it is *minimax optimal* in the sense that it matches the lower bound we previously derived. Unlike most prior work, we provide non-asymptotic performance guarantees which hold even for a finite number of workers and tasks. As our analysis reveals, non-asymptotic performance guarantees require to use finer concentration arguments than asymptotic ones.

TE has low complexity in terms of memory space and computation time, does not require to store the whole data set in memory and can be easily applied in a setting in which answers to tasks arrive

sequentially, i.e., in a *streaming* setting. Finally, we compare the performance of TE to state-of-the-art algorithms through numerical experiments using both synthetic and real datasets.

## 2   Related Work

The first problems of data classification using independent workers appeared in the medical context, where each label refers to the state of a patient (e.g., sick or sane) and the workers are clinicians. [Dawid and Skene, 1979] proposed an expectation-maximization (EM) algorithm, admitting that the accuracy of the estimate was unknown. Several versions and extensions of this algorithm have since been proposed and tested in various settings [Hui and Walter, 1980, Smyth et al., 1995, Albert and Dodd, 2004, Raykar et al., 2010, Liu et al., 2012].

A number of Bayesian techniques have also been proposed and applied to this problem by [Raykar et al., 2010, Welinder and Perona, 2010, Karger et al., 2011, Liu et al., 2012, Karger et al., 2014, 2013] and references therein. Of particular interest is the belief-propagation (BP) algorithm of [Karger et al., 2011], which is provably order-optimal in terms of the number of workers required per task for any given target error rate, in the limit of an infinite number of tasks and an infinite population of workers.

Another family of algorithms is based on the spectral analysis of some matrix representing the correlations between tasks or workers. [Ghosh et al., 2011] work on the task-task matrix whose entries correspond to the number of workers having labeled two tasks in the same manner, while [Dalvi et al., 2013] work on the worker-worker matrix whose entries correspond to the number of tasks labeled in the same manner by two workers. Both obtain performance guarantees by the perturbation analysis of the top eigenvector of the corresponding expected matrix. The BP algorithm of Karger, Oh and Shah is in fact closely related to these spectral algorithms: their message-passing scheme is very similar to the power-iteration method applied to the task-worker matrix, as observed in [Karger et al., 2011].

Two notable recent contributions are [Chao and Dengyong, 2015] and [Zhang et al., 2014]. The former provides performance guarantees for two versions of EM, and derives lower bounds on the attainable prediction error (the probability of estimating labels incorrectly). The latter provides lower bounds on the estimation error of the workers' reliability as well as performance guarantees for an improved version of EM relying on spectral methods in the initialization phase. Our lower bound cannot be compared to that of [Chao and Dengyong, 2015] because it applies to the workers' reliability and not the prediction error; and our lower bound is tighter than that of [Zhang et al., 2014]. Our estimator shares some features of the algorithm proposed by [Zhang et al., 2014] to initialize EM, which suggests that the EM phase itself is not essential to attain minimax optimality.

All these algorithms require the storage of all labels in memory and, to the best of our knowledge, the only known streaming algorithm is the recursive EM algorithm of [Wang et al., 2013], for which no performance guarantees are available.

The remainder of the paper is organized as follows. In section 3 we state the problem and introduce our notations. The important question of identifiability is addressed in section 4. In section 5 we present a lower bound on the minimax error rate of any estimator. In section 6 we present TE, discuss its compexity and prove that it is minimax optimal. In section 7 we present numerical experiments on synthetic and real-world data sets and section 8 concludes the paper. Due to space constraints, we only provide proof outlines for our two main results in this document. Complete proofs are presented in the supplementary material.

## 3   Model

Consider $n$ workers, for some integer $n \geq 3$. Each task consists in determining the answer to a binary question. The answer to task $t$, the "ground-truth", is denoted by $G(t) \in \{+1, -1\}$. We assume that the random variables $G(1), G(2), \ldots$ are i.i.d. and centered, so that there is no bias towards one of the answers.

Each worker provides an answer with probability $\alpha \in (0, 1]$. When worker $i \in \{1, ..., n\}$ provides an answer, this answer is correct with probability $\frac{1}{2}(1 + \theta_i)$, independently of the other workers, for some parameter $\theta_i \in [-1, 1]$ that we refer to as the *reliability* of worker $i$. If $\theta_i > 0$ then worker

$i$ tends to provide correct answers; if $\theta_i < 0$ then worker $i$ tends to provide incorrect anwsers; if $\theta_i = 0$ then worker $i$ is non-informative. We denote by $\theta = (\theta_1, \ldots, \theta_n)$ the reliability vector. Both $\alpha$ and $\theta$ are unknown.

Let $X_i(t) \in \{-1, 0, 1\}$ be the output of worker $i$ for task $t$, where the output 0 corresponds to the absence of an answer. We have:

$$X_i(t) = \begin{cases} G(t) & \text{w.p.} & \alpha \frac{1+\theta_i}{2}, \\ -G(t) & \text{w.p.} & \alpha \frac{1-\theta_i}{2} \\ 0 & \text{w.p.} & 1-\alpha. \end{cases} \tag{1}$$

Since the workers are independent, the random variables $X_1(t), ..., X_n(t)$ are independent given $G(t)$, for each task $t$. We denote by $X(t)$ the corresponding vector. The goal is to estimate the ground-truth $G(t)$ as accurately as possible by designing an estimator $\hat{G}(t)$ that minimizes the error probability $\mathbb{P}(\hat{G}(t) \neq G(t))$. The estimator $\hat{G}(t)$ is adaptive and may be a function of $X(1), ..., X(t)$ but not of the unknown parameters $\alpha, \theta$.

It is well-known that, given $\theta$ and $\alpha = 1$, an optimal estimator of $G(t)$ is the weighted majority vote [Nitzan and Paroush, 1982, Shapley and Grofman, 1984], namely

$$\hat{G}(t) = \mathbf{1}\{W(t) > 0\} - \mathbf{1}\{W(t) < 0\} + Z\mathbf{1}\{W(t) = 0\}, \tag{2}$$

where $W(t) = \frac{1}{n}\sum_{i=1}^n w_i X_i(t)$, $w_i = \ln(\frac{1+\theta_i}{1-\theta_i})$ is the weight of worker $i$ (possibly infinite), and $Z$ is a Bernoulli random variable of parameter $\frac{1}{2}$ over $\{+1, -1\}$ (for random tie-breaking). We prove this result for any $\alpha \in (0, 1]$.

**Proposition 1** *Assuming that $\theta$ is known, the estimator* (2) *is an optimal estimator of $G(t)$.*

*Proof.* Finding an optimal estimator of $G(t)$ amounts to finding an optimal statistical test between hypotheses $\{G(t) = +1\}$ and $\{G(t) = -1\}$, under a symmetry constraint so that type I and type II error probability are equal. For any $x \in \{-1, 0, 1\}^n$, let $L^+(x)$ and $L^-(x)$ be the probabilities that $X(t) = x$ under hypotheses $\{G(t) = +1\}$ and $\{G(t) = -1\}$, respectively. We have

$$L^+(x) = H(x)\prod_{i=1}^n (1+\theta_i)^{\mathbf{1}\{x_i=+1\}}(1-\theta_i)^{\mathbf{1}\{x_i=-1\}},$$

$$L^-(x) = H(x)\prod_{i=1}^n (1+\theta_i)^{\mathbf{1}\{x_i=-1\}}(1-\theta_i)^{\mathbf{1}\{x_i=+1\}},$$

where $\ell = \sum_{i=1}^n |x_i|$ is the number of answers and $H(x) = \frac{1}{2^\ell}\alpha^\ell(1-\alpha)^{n-\ell}$. We deduce the log-likelihood ratio $\ln\left(\frac{L^+(x)}{L^-(x)}\right) = \sum_{i=1}^n w_i x_i$. By the Neyman-Pearson theorem, for any level of significance, there exists $a$ and $b$ such that the uniformly most powerful test for that level is: $\mathbf{1}\{w^T x > a\} - \mathbf{1}\{w^T x < a\} + Z\mathbf{1}\{w^T x = a\}$, where $Z$ is a Bernoulli random variable of parameter $b$ over $\{+1, -1\}$. By symmetry, we must have $a = 0$ and $b = \frac{1}{2}$, as announced. $\square$

This result shows that estimating the true answer $G(t)$ reduces to estimating the unknown parameters $\alpha$ and $\theta$, which is the focus of the paper. Note that the problem of estimating $\theta$ is important in itself, due to the presence of "spammers" (i.e., workers with low reliability); a good estimator can be used by the crowdsourcing platform to incentivize good workers.

## 4 Identifiability

Estimating $\alpha$ and $\theta$ from $X(1), ..., X(t)$ is not possible unless we have *identifiability*, namely there cannot exist two distinct sets of parameters $\alpha, \theta$ and $\alpha', \theta'$ under which the distribution of $X(1), ..., X(t)$ is the same. Let $X \in \{-1, 0, 1\}^n$ be any sample, for some parameters $\alpha \in (0, 1]$ and $\theta \in [-1, 1]^n$. The parameter $\alpha$ is clearly identifiable since $\alpha = \mathbb{P}(X_1 \neq 0)$. The identifiability of $\theta$ is less obvious. Assume for instance that $\theta_i = 0$ for all $i \geq 3$. It follows from (1) that for any $x \in \{-1, 0, 1\}^n$, with $H(x)$ defined as in the proof of Proposition 1:

$$\mathbb{P}(X = x) = H(x) \times \begin{cases} 1+\theta_1\theta_2 & \text{if } x_1 x_2 = 1, \\ 1-\theta_1\theta_2 & \text{if } x_1 x_2 = -1, \\ 1 & \text{if } x_1 x_2 = 0. \end{cases}$$

In particular, two parameters $\theta, \theta'$ such that $\theta_1 \theta_2 = \theta'_1 \theta'_2$ and $\theta_i = \theta'_i = 0$ for all $i \geq 3$ cannot be distinguished. Similarly, by symmetry, two parameters $\theta, \theta'$ such that $\theta' = -\theta$ cannot be distinguished. Let:

$$\Theta = \left\{ \theta \in [-1,1]^n : \sum_{i=1}^n \mathbf{1}\{\theta_i \neq 0\} \geq 3, \sum_{i=1}^n \theta_i > 0 \right\}.$$

The first condition states that there are at least 3 informative workers, the second that the average reliability is positive.

**Proposition 2** *Any parameter $\theta \in \Theta$ is identifiable.*

*Proof.* Any parameter $\theta \in \Theta$ can be expressed as a function of the covariance matrix of $X$ (section 6 below): the absolute value and the sign of $\theta$ follow from (4) and (5), respectively. $\qquad\square$

## 5 Lower bound on the minimax error

The estimation of $\alpha$ is straightforward and we here focus on the best estimation of $\theta$ one can expect, assuming $\alpha$ is known. Specifically, we derive a lower bound on the minimax error of any estimator $\hat{\theta}$ of $\theta$. Define $||\hat{\theta} - \theta||_\infty = \max_{i=1,\ldots,n} |\hat{\theta}_i - \theta_i|$ and for all $\theta \in [-1,1]^n$, $A(\theta) = \min_k \max_{i,j \neq k} \sqrt{|\theta_i \theta_j|}$ and $B(\theta) = \sum_{i=1}^n \theta_i$.

Observe that $\Theta = \{\theta \in [-1,1]^n : A(\theta) > 0, \ B(\theta) > 0\}$. This suggests that the estimation of $\theta$ becomes hard when either $A(\theta)$ or $B(\theta)$ is small. Define for any $a, b \in (0,1)$, $\Theta_{a,b} = \{\theta \in [-1,1]^n : A(\theta) \geq a, \ B(\theta) \geq b\}$. We have the following lower bound on the minimax error. As the proof reveals, the parameters $a$ and $b$ characterize the difficulty of estimating the absolute value and the sign of $\theta$, respectively.

**Theorem 1 (Minimax error)** *Consider any estimator $\hat{\theta}(t)$ of $\theta$.*

*For any $\epsilon \in (0, \min(a, (1-a)/2, 1/4))$ and $\delta \in (0, 1/4)$, we have*

$$\min_{\theta \in \Theta_{a,b}} \mathbb{P}\left( ||\hat{\theta}(t) - \theta||_\infty \geq \epsilon \right) \geq \delta \,, \ \forall t \leq \max(T_1, T_2),$$

*with $T_1 = c_1 \frac{1-a}{\alpha^2 a^4 \epsilon^2} \ln\left(\frac{1}{4\delta}\right)$, $T_2 = c_2 \frac{(1-a)^4 (n-4)}{\alpha^2 a^2 b^2} \ln\left(\frac{1}{4\delta}\right)$ and $c_1, c_2 > 0$ two universal constants.*

**Outline of proof.** The proof is based on an information theoretic argument. Denote by $P_\theta$ the distribution of $X$ under parameter $\theta \in \Theta$, and $D(.||.)$ the Kullback-Leibler (KL) divergence. The main element of proof is lemma 1, where we bound $D(P_{\theta'}||P_\theta)$ for two well chosen pairs of parameters. The pair $\theta, \theta'$ in statement (i) is hard to distinguish when $a$ is small, hence it is hard to estimate the absolute value of $\theta$. The pair $\theta, \theta'$ of statement (ii) is also hard to distinguish when $a$ or $b$ are small, which shows that it is difficult to estimate the sign of $\theta$. Proving lemma 1 is involved because of the particular form of distribution $P_\theta$, and requires careful manipulations of the likelihood ratio. We conclude by reduction to a binary hypothesis test between $\theta$ and $\theta'$ using lemma 2.

**Lemma 1** *(i) Let $a \in (0,1)$, $\theta = (1, a, a, 0, \ldots, 0)$ and $\theta' = (1 - 2\epsilon, \frac{a}{1-2\epsilon}, \frac{a}{1-2\epsilon}, 0, \ldots, 0)$.*

*Then: $D(P_{\theta'}||P_\theta) \leq \frac{1}{c_1} \frac{\alpha^2 a^4 \epsilon^2}{1-a}$ (ii) Let $n > 4$, define $c = b/(n-4)$, and $\theta = (a, a, -a, -a, c, \ldots, c), \theta' = (-a, -a, a, a, c, \ldots, c)$. Then: $D(P_{\theta'}||P_\theta) \leq \frac{1}{c_2} \frac{\alpha^2 a^2 b^2}{(n-4)(1-a)^4}$.*

**Lemma 2** *[Tsybakov, 2008, Theorem 2.2] Consider any estimator $\hat{\theta}(t)$.*

*For any $\theta, \theta' \in \Theta$ with $||\theta - \theta'||_\infty \geq 2\epsilon$ we have:*

$$\min\left( \mathbb{P}_\theta(||\hat{\theta}(t) - \theta||_\infty \geq \epsilon), \mathbb{P}_{\theta'}(||\hat{\theta}(t) - \theta'||_\infty \geq \epsilon) \right) \geq \frac{1}{4 \exp(-tD(P_{\theta'}||P_\theta))}.$$

**Relation with prior work.** The lower bound derived in [Zhang et al., 2014][Theorem 3] shows that the minimax error of any estimator $\hat{\theta}$ must be greater than $\mathcal{O}((\alpha t)^{-\frac{1}{2}})$. Our lower bound is stricter, and shows that the minimax error is in fact greater than $\mathcal{O}(a^{-2}\alpha^{-1}t^{-\frac{1}{2}})$. Another lower bound was derived in [Chao and Dengyong, 2015][Theorems 3.4 and 3.5], but this concerns the prediction error rate, that is $\mathbb{P}(\hat{G} \neq G)$, so that it cannot be easily compared to our result.

# 6 Triangular estimation

We here present our estimator. The absolute value of the reliability of each worker $k$ is estimated through the correlation of her answers with those of the most informative pair $i, j \neq k$. We refer to this algorithm as *triangular estimation* (TE). The sign of the reliability of each worker is estimated in a second step. We use the convention that $\text{sign}(0) = +$.

**Covariance matrix.** Let $X \in \{-1, 0, 1\}^n$ be any sample, for some parameters $\alpha \in (0, 1]$ and $\theta \in \Theta$. We shall see that the parameter $\theta$ could be recovered exactly if the covariance matrix of $X$ were perfectly known. For any $i \neq j$, let $C_{ij}$ be the covariance of $X_i$ and $X_j$ given that $X_i X_j \neq 0$ (that is, both workers $i$ and $j$ provide an answer). In view of (1),

$$C_{ij} = \frac{\mathbb{E}(X_i X_j)}{\mathbb{E}(|X_i X_j|)} = \theta_i \theta_j. \tag{3}$$

In particular, for any distinct indices $i, j, k$, $C_{ik} C_{jk} = \theta_i \theta_j \theta_k^2 = C_{ij} \theta_k^2$. We deduce that, for any $k = 1, \ldots, n$ and any pair $i, j \neq k$ such that $C_{ij} \neq 0$,

$$\theta_k^2 = \frac{C_{ik} C_{jk}}{C_{ij}}. \tag{4}$$

Note that such a pair exists for each $k$ because $\theta \in \Theta$. To recover the sign of $\theta_k$, we use the fact that $\theta_k \sum_{i=1}^n \theta_i = \theta_k^2 + \sum_{i \neq k} C_{ik}$. Since $\theta \in \Theta$, we get

$$\text{sign}(\theta_k) = \text{sign}\left(\theta_k^2 + \sum_{i \neq k} C_{ik}\right). \tag{5}$$

The TE algorithm consists in estimating the covariance matrix to recover $\theta$ from the above expressions.

**TE algorithm.** At any time $t$, define

$$\forall i, j = 1, \ldots, n, \quad \hat{C}_{ij} = \frac{\sum_{s=1}^t X_i(s) X_j(s)}{\max\left(\sum_{s=1}^t |X_i(s) X_j(s)|, 1\right)}. \tag{6}$$

For all $k = 1, \ldots, n$, find the most informative pair $(i_k, j_k) \in \arg\max_{i \neq j \neq k} |\hat{C}_{ij}|$ and let

$$|\hat{\theta}_k| = \begin{cases} \sqrt{\left|\frac{\hat{C}_{i_k k} \hat{C}_{j_k k}}{\hat{C}_{i_k j_k}}\right|} & \text{if } |\hat{C}_{i_k j_k}(t)| > 0, \\ 0 & \text{otherwise.} \end{cases}$$

Next, define $k^* = \arg\max_k \left|\hat{\theta}_k^2 + \sum_{i \neq k} \hat{C}_{ik}\right|$ and let

$$\text{sign}(\hat{\theta}_k) = \begin{cases} \text{sign}(\hat{\theta}_{k^*}^2 + \sum_{i \neq k^*} \hat{C}_{ik^*}) & \text{if } k = k^*, \\ \text{sign}(\hat{\theta}_{k^*} \hat{C}_{kk^*}) & \text{otherwise,} \end{cases}$$

**Complexity.** First note that the TE algorithm is a streaming algorithm since $\hat{C}_{ij}(t)$ can be written

$$\hat{C}_{ij} = \frac{M_{ij}}{\max(N_{ij}, 1)} \text{ with } M_{ij} = \sum_{s=1}^t X_i(s) X_j(s) \text{ and } N_{ij} = \sum_{s=1}^t |X_i(s) X_j(s)|.$$

Thus TE requires $\mathcal{O}(n^2)$ memory space (to store the matrices $M$ and $N$) and has a time complexity of $\mathcal{O}(n^2 \ln(n))$ per task: $\mathcal{O}(n^2)$ operations to update $\hat{C}$, $\mathcal{O}(n^2 \ln(n))$ operations to sort the entries of $|\hat{C}(t)|$, $\mathcal{O}(n^2)$ operations to compute $|\hat{\theta}|$, $\mathcal{O}(n^2)$ operations to compute the sign of $\hat{\theta}$.

**Minimax optimality.** The following result shows that the proposed estimator is minimax optimal. Namely the sample complexity of our estimator matches the lower bound up to an additive logarithmic term $\ln(n)$ and a multiplicative constant.

**Theorem 2** *Let $\theta \in \Theta_{a,b}$ and denote by $\hat{\theta}(t)$ the estimator defined above. For any $\epsilon \in (0, \min(\frac{b}{3}, 1))$ and $\delta \in (0, 1)$, we have*

$$\mathbb{P}(||\hat{\theta}(t) - \theta||_\infty \geq \epsilon) \leq \delta \quad , \quad \forall t \geq \max(T_1', T_2'),$$

*with $T_1' = c_1' \frac{1}{\alpha^2 a^4 \epsilon^2} \ln\left(\frac{6n^2}{\delta}\right)$, $T_2' = c_2' \frac{n}{\alpha^2 a^2 b^2} \ln\left(\frac{4n^2}{\delta}\right)$, and $c_1', c_2' > 0$ two universal constants.*

**Outline of proof.** Define $||\hat{C} - C||_\infty = \max_{i,j:i\neq j} |\hat{C}_{ij} - C_{ij}|$. The TE estimator is a function of the empirical pairwise correlations $(\hat{C}_{ij})_{i,j}$ and the sums $\sum_{j\neq i} \hat{C}_{ij}$. The main difficulty is to prove lemma 3, a concentration inequality for $\sum_{j\neq i} \hat{C}_{ij}$.

**Lemma 3** *For all $i = 1, \ldots, n$ and all $\varepsilon > 0$,*

$$\mathbb{P}\left(|\sum_{j\neq i}(\hat{C}_{ij} - C_{ij})| \geq \varepsilon\right) \leq 2\exp\left(-\frac{\varepsilon^2 \alpha^2 t}{30 \max(B(\theta)^2, n)}\right) + 2n\exp\left(-\frac{t\alpha^2}{8(n-1)}\right).$$

Consider $i$ fixed. We dissociate the set of tasks answered by each worker from the actual answers and the truth. Let $U = (U_j(t))_{j,t}$ be i.i.d Bernoulli random variables with $\mathbb{E}(U_j(t)) = \alpha$ and $V = (V_j(t))_{j,t}$ be independent random variables on $\{-1, 1\}$ with $\mathbb{E}(V_j(t)) = \theta_j$. One may readily check that $(X_j(t))_{j,t}$ has the same distribution as $(G(t)U_j(t)V_j(t))_{j,t}$. Hence, in distribution:

$$\sum_{j\neq i}\hat{C}_{ij} = \sum_{j\neq i}\sum_{s=1}^{t}\frac{U_i(s)U_j(s)V_i(s)V_j(s)}{N_j} \text{ with } N_j = \sum_{s=1}^{t}U_i(s)U_j(s).$$

We prove lemma 3 by conditionning with respect to $U$. Denote by $\mathbb{P}_U$ the conditional probability with respect to $U$. Define $N = \min_{j\neq i} N_{ij}$. We prove that for all $\varepsilon \geq 0$:

$$\mathbb{P}_U\left(\sum_{j\neq i}(\hat{C}_{ij} - C_{ij}) \geq \varepsilon\right) \leq e^{-\frac{\varepsilon^2}{\sigma^2}} \text{ with } S = \sum_{s=1}^{t}\left(\sum_{j\neq i}U_i(s)U_j(s)\theta_j\right)^2 \text{ and } \sigma^2 = \frac{(n-1)N + S}{N^2}.$$

The quantity $\sigma$ is an upper bound on the conditional variance of $\sum_{j\neq i}\hat{C}_{ij}$, which we control by applying Chernoff's inequality to both $N$ and $S$. We get:

$$\mathbb{P}(N \leq \alpha^2 t/2) \leq (n-1)e^{-\frac{t\alpha^2}{8}} \quad \text{and} \quad \mathbb{P}(S \geq 2t\alpha^2 \max(B_i(\theta)^2, n-1)) \leq e^{-\frac{t\alpha^2}{3(n-1)}}.$$

Removing the conditionning on $U$ yields the result. We conclude the proof of theorem 2 by linking the fluctuations of $\hat{C}$ to that of $\hat{\theta}$ in lemma 4.

**Lemma 4** *If (a) $||\hat{C} - C||_\infty \leq \varepsilon \leq A^2(\theta) \min(\frac{1}{2}, \frac{B(\theta)}{64})$ and (b) $\max_i |\sum_{j\neq i}\hat{C}_{ij} - C_{ij}| \leq \frac{A(\theta)B(\theta)}{8}$, then $||\hat{\theta} - \theta||_\infty \leq \frac{24\varepsilon}{A^2(\theta)}$.*

**Relation with prior work.** Our upper bound brings improvement over [Zhang et al., 2014] as follows. Two conditions are required for the upper bound of [Zhang et al., 2014][Theorem 4] to hold: (i) it is required that $\max_i |\theta_i| < 1$, and (ii) the number of workers $n$ must grow with both $\delta$ and $t$, and in fact must depend on $a$ and $b$, so that $n$ has to be large if $b$ is smaller than $\sqrt{n}$. Our result does not require condition (i) to hold. Further there are values of $a$ and $b$ such that condition (ii) is never satisfied, for instance $n \geq 5$, $a = \frac{1}{2}$, $b = \frac{\sqrt{n-4}}{2}$ and $\theta = (a, -a, a, -a, \frac{b}{n-4}, \ldots, \frac{b}{n-4}) \in \Theta_{a,b}$. For [Zhang et al., 2014][Theorem 4] to hold, $n$ should satisfy $n \geq c_3 n \ln(t^2 n/\delta)$ with $c_3$ a universal constant (see discussion in the supplement) and for $t$ or $1/\delta$ large enough no such $n$ exists. It is noted that for such values of $a$ and $b$, our result remains informative. Our result shows that one can obtain a minimax optimal algorithm for crowdsourcing which does not involve any EM step.

The analysis of [Chao and Dengyong, 2015] also imposes $n$ to grow with $t$ and conditions on the minimal value of $b$. Specifically the first and the last condition of [Chao and Dengyong, 2015][Theorem

3.3], require that $n \geq \ln(t)$ and that $\sum_i \theta_i^2 \geq 6\ln(t)$. Using the previous example (even for $t = 3$), this translates to $b \geq 2\sqrt{n - 4}$.

In fact, the value $b = \mathcal{O}(\sqrt{n})$ seems to mark the transition between "easy" and "hard" instances of the crowdsourcing problem. Indeed, when $n$ is large and $b$ is large with respect to $\sqrt{n}$, then the majority vote outputs the truth with high probability by the Central Limit Theorem.

## 7 Numerical Experiments

**Synthetic data.** We consider three instances: (i) $n = 50$, $t = 10^3$, $\alpha = 0.25$, $\theta_i = a$ if $i \leq n/2$ and 0 otherwise; (ii) $n = 50$, $t = 10^4$, $\alpha = 0.25$, $\theta = (1, a, a, 0, ..., 0)$; (iii) $n = 50$, $t = 10^4$, $\alpha = 0.25$, $a = 0.9$, $\theta = (a, -a, a, -a, \frac{b}{n-4}, ..., \frac{b}{n-4})$.

Instance (i) is an "easy" instance where half of the workers are informative, with $A(\theta) = a$ and $B(\theta) = na/2$. Instance (ii) is a "hard" instance, the difficulty being to estimate the absolute value of $\theta$ accurately by identifying the 3 informative workers. Instance (iii) is another "hard" instance, where estimating the sign of the components of $\theta$ is difficult. In particular, one must distinguish $\theta$ from $\theta' = (-a, a, -a, a, \frac{b}{n-4}, ..., \frac{b}{n-4})$, otherwise a large error occurs.

Both "hard" instances (ii) and (iii) are inspired by our derivation of the lower bound and constitute the hardest instances in $\Theta_{a,b}$. For each instance we average the performance of algorithms on $10^3$ independent runs and apply a random permutation of the components of $\theta$ before each run. We consider the following algorithms: KOS (the BP algorithm of [Karger et al., 2011]), Maj (majority voting), Oracle (weighted majority voting with optimal weights, the optimal estimator of the ground truth), RoE (first spectral algorithm of [Dalvi et al., 2013]), EoR (second spectral algorithm of [Dalvi et al., 2013]), GKM (spectral algorithm of [Ghosh et al., 2011]), S-EM$k$ (EM algorithm with spectral initialization of [Zhang et al., 2014] with $k$ iterations of EM) and TE (our algorithm). We do not present the estimation error of KOS, Maj and Oracle since these algorithms only predict the ground truth but do not estimate $\theta$ directly.

The results are shown in Tables 1 and 2, where the best results are indicated in bold. The spectral algorithms RoE, EoR and GKM tend to be outperformed by the other algorithms. To perform well, GKM needs $\theta_1$ to be positive and large (see [Ghosh et al., 2011]); whenever $\theta_1 \leq 0$ or $|\theta_1|$ is small, GKN tends to make a sign mistake causing a large error. Also the analysis of RoE and EoR assumes that the task-worker graph is a random $D$-regular graph (so that the worker-worker matrix has a large spectral gap). Here this assumption is violated and the practical performance suffers noticeably, so that this limitation is not only theoretical. KOS performs consistently well, and seems immune to sign ambiguity, see instance (iii). Further, while the analysis of KOS also assumes that the task-worker graph is random $D$-regular, its practical performance does not seem sensitive to that assumption. The performance of S-EM is good except when sign estimation is hard (instance (iii), $b = 1$). This seems due to the fact that the initialization of S-EM (see the algorithm description) is not good in this case. Hence the limitation of $b$ being of order $\sqrt{n}$ is not only theoretical but practical as well. In fact (combining our results and the ideas of [Zhang et al., 2014]), this suggests a new algorithm where one uses EM with TE as the initial value of $\theta$.

Further, the number of iterations of EM brings significant gains in some cases and should affect the universal constants in front of the various error bounds (providing theoretical evidence for this seems non trivial). TE performs consistently well except for (i) $a = 0.3$ (which we believe is due to the fact that $t$ is relatively small in that instance). In particular when sign estimation is hard TE clearly outperforms the competing algorithms. This indeed suggests two regimes for sign estimation: $b = \mathcal{O}(1)$ (hard regime) and $b = \mathcal{O}(\sqrt{n})$ (easy regime).

**Real-world data.** We next consider 6 publicly available data sets (see [Whitehill et al., 2009, Zhou et al., 2015] and summary information in Table 3), each consisting of labels provided by workers and the ground truth. The density is the average number of labels per worker, i.e., $\alpha$ in our model. The worker degree is the average number of tasks labeled by a worker.

First, for data sets with more than 2 possible label values, we split the label values into two groups and associate them with $-1$ and $+1$ respectively. The partition of the labels is given in Table 3. Second, we remove any worker who provides less than 10 labels. Our preliminary numerical experiments (not shown here for concision) show that without this, none of the studied algorithms

even match the majority consistently. Workers with low degree create noise and (to the best of our knowledge) any theoretical analysis of crowdsourcing algorithms assumes that the worker degree is sufficiently large. The performance of various algorithms is reported in Table 4. No information about the workers reliability is available so we only report the prediction error $\mathbb{P}(\hat{G} \neq G)$. Further, one cannot compare algorithms to the Oracle, so that the main goal is to outperform the majority.

Apart from "Bird" and "Web", none of the algorithms seem to be able to significantly outperform the majority and are sometimes noticeably worse. For "Web" which has both the largest number of labels and a high worker degree, there is a significant gain over the majority vote, and TE, despite its low complexity, slightly outperforms S-EM and is competitive with KOS and GKM which both perform best on this dataset.

| Instance | RoE | EoR | GKM | S-EM1 | S-EM10 | TE |
|---|---|---|---|---|---|---|
| (i) $a = 0.3$ | 0.200 | 0.131 | 0.146 | 0.100 | **0.041** | 0.134 |
| (i) $a = 0.9$ | 0.274 | 0.265 | 0.271 | **0.022** | **0.022** | 0.038 |
| (ii) $a = 0.55$ | 0.551 | 0.459 | 0.479 | 0.045 | **0.044** | 0.050 |
| (ii) $a = 0.95$ | 0.528 | 0.522 | 0.541 | 0.034 | **0.033** | 0.039 |
| (iii) $b = 1$ | 0.253 | 0.222 | 0.256 | 0.533 | 0.389 | **0.061** |
| (iii) $b = \sqrt{n}$ | 0.105 | 0.075 | 0.085 | 0.437 | **0.030** | 0.045 |

Table 1: Synthetic data: estimation error $\mathbb{E}(||\hat{\theta} - \theta||_\infty)$.

| Instance | **Oracle** | Maj | KOS | RoE | EoR | GKM | S-EM1 | S-EM10 | TE |
|---|---|---|---|---|---|---|---|---|---|
| (i) $a = 0.3$ | 0.227 | 0.298 | 0.228 | 0.402 | 0.398 | 0.374 | 0.251 | **0.228** | 0.250 |
| (i) $a = 0.9$ | 0.004 | 0.046 | 0.004 | 0.217 | 0.218 | 0.202 | **0.004** | **0.004** | **0.004** |
| (ii) $a = 0.55$ | 0.284 | 0.441 | 0.292 | 0.496 | 0.497 | 0.495 | **0.284** | 0.285 | **0.284** |
| (ii) $a = 0.95$ | 0.219 | 0.419 | 0.220 | 0.495 | 0.496 | 0.483 | **0.219** | **0.219** | **0.219** |
| (iii) $b = 1$ | 0.181 | 0.472 | **0.185** | 0.443 | 0.455 | 0.386 | 0.388 | 0.404 | 0.192 |
| (iii) $b = \sqrt{n}$ | 0.126 | 0.315 | 0.133 | 0.266 | 0.284 | 0.207 | 0.258 | **0.127** | 0.128 |

Table 2: Synthetic data: prediction error $\mathbb{P}(\hat{G} \neq G)$.

| Data Set | # Tasks | # Workers | # Labels | Density | Worker Degree | Label Domain |
|---|---|---|---|---|---|---|
| Bird | 108 | 39 | 4,212 | 1 | 108 | {0} vs {1} |
| Dog | 807 | 109 | 8,070 | 0.09 | 74 | {0,2} vs {1,3} |
| Duchenne | 159 | 64 | 1,221 | 0.12 | 19 | {0} vs {1} |
| RTE | 800 | 164 | 8,000 | 0.06 | 49 | {0} vs {1} |
| Temp | 462 | 76 | 4,620 | 0.13 | 61 | {1} vs {2} |
| Web | 2,653 | 177 | 15,539 | 0.03 | 88 | {1,2,3} vs {4,5} |

Table 3: Summary of the real-world datasets.

| Data Set | **Maj** | KOS | RoE | EoR | GKM | S-EM1 | S-EM10 | TE |
|---|---|---|---|---|---|---|---|---|
| Bird | 0.24 | 0.28 | 0.29 | 0.29 | 0.28 | 0.20 | 0.28 | **0.18** |
| Dog | 0.18 | 0.19 | 0.18 | 0.18 | 0.20 | 0.24 | **0.17** | 0.20 |
| Duchenne | 0.28 | 0.30 | 0.29 | 0.28 | 0.29 | 0.28 | 0.30 | **0.26** |
| RTE | **0.10** | 0.50 | 0.50 | 0.89 | 0.49 | 0.32 | 0.16 | 0.38 |
| Temp | **0.06** | 0.43 | 0.24 | 0.10 | 0.43 | **0.06** | **0.06** | 0.08 |
| Web | 0.14 | **0.02** | 0.13 | 0.14 | **0.02** | 0.04 | 0.06 | 0.03 |

Table 4: Real-world data: prediction error $\mathbb{P}(\hat{G} \neq G)$.

# 8 Conclusion

We have derived a minimax error lower bound for the crowdsourcing problem and have proposed TE, a low-complexity algorithm which matches this lower bound. Our results open several questions of interest. First, while recent work has shown that one can obtain strong theoretical guarantees by combining one step of EM with a well-chosen initialization, we have shown that, at least in the case of binary labels, one can forgo the EM phase altogether and still obtain both minimax optimality and good numerical performance. It would be interesting to know if this is still possible when there are more than two possible labels, and also if one can do so using a streaming algorithm.

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
