[Supplementary Material]

# A Minimax Optimal Algorithm for Crowdsourcing Supplementary Material

**Thomas Bonald**
Telecom ParisTech
thomas.bonald@telecom-paristech.fr

**Richard Combes**
Centrale-Supelec / L2S
richard.combes@supelec.fr

## Abstract

We here provide the proofs of the two main results of the paper, and provide a more in-depth discussion on the relation between our upper bound and that of [Zhang et al., 2014].

## 1 Proof of Theorem 1

We use the following inequality between the Kullback-Leibler and $\chi^2$ divergences.

**Lemma 1** *The Kullback-Leibler divergence of any two discrete distributions $P, Q$ satisfies*

$$D(P||Q) \leq \mathbb{E}\left(\frac{(P(X) - Q(X))^2}{P(X)Q(X)}\right),$$

*with $X \sim P$.*

*Proof.* Using the inequality $\ln(z) \leq z - 1$, we get

$$D(P||Q) = \sum_x P(x)\ln\left(\frac{P(x)}{Q(x)}\right) \leq \sum_x P(x)\left(\frac{P(x)}{Q(x)} - 1\right) = -1 + \sum_x \frac{P(x)^2}{Q(x)}.$$

Writing

$$P(x)^2 = Q(x)^2 + 2Q(x)(P(x) - Q(x)) + (P(x) - Q(x))^2,$$

we deduce

$$
\begin{aligned}
D(P||Q) &\leq -1 + \sum_x Q(x) + 2\sum_x (P(x) - Q(x)) + \sum_x \frac{(P(x) - Q(x))^2}{Q(x)}, \\
&= \sum_x \frac{(P(x) - Q(x))^2}{Q(x)}, \\
&= \sum_x P(x)\frac{(P(x) - Q(x))^2}{P(x)Q(x)}.
\end{aligned}
$$

$\square$

**Proof of Theorem 1.** Let $X \in \{+1, 0, -1\}^n$ be any sample under parameters $\alpha, \theta$. We have for any distinct indices $i, j, k \in \{1, \ldots, n\}$,

$$\mathbb{P}((X_i, X_j, X_k) = (0, 1, 1)) = \mathbb{P}((X_i, X_j, X_k) = (0, -1, -1)) = (1 - \alpha)\frac{\alpha^2}{4}(1 + \theta_j\theta_k),$$

$$\mathbb{P}((X_i, X_j, X_k) = (0, 1, -1)) = \mathbb{P}((X_i, X_j, X_k) = (0, -1, 1)) = (1 - \alpha)\frac{\alpha^2}{4}(1 - \theta_j\theta_k),$$

$$\mathbb{P}((X_i, X_j, X_k) = (1, 1, 1)) = \mathbb{P}((X_i, X_j, X_k) = (-1, -1, -1)) = \frac{\alpha^3}{8}(1 + \theta_i\theta_j + \theta_j\theta_k + \theta_k\theta_i),$$

$$\mathbb{P}((X_i, X_j, X_k) = (1, 1, -1)) = \mathbb{P}((X_i, X_j, X_k) = (-1, -1, 1)) = \frac{\alpha^3}{8}(1 + \theta_i\theta_j - \theta_j\theta_k - \theta_k\theta_i).$$

Now let $a \in (0, 1)$ and

$$\theta = (1, a, a, 0, \ldots, 0), \quad \theta' = (1 - 2\epsilon, \frac{a}{1 - 2\epsilon}, \frac{a}{1 - 2\epsilon}, 0, \ldots, 0).$$

Observe that $\theta, \theta' \in \Theta_{a,b}$ for any $b \in (0, 1)$. Denote by $P, P'$ the distributions of $X$ under parameters $\theta, \theta'$, respectively. We use Lemma 1 to get an upper bound on the Kullback-Leibler divergence $D(P'||P)$ between $P'$ and $P$. Observe that we can restrict the analysis to the case $n = 3$. We calculate $P(x), P'(x)$ for all possible values of $x \in \{-1, 0, 1\}^3$:

(a) If $x_2 = 0$ or $x_3 = 0$ then $P(x) = P'(x)$.

(b) If $x = (0, 1, 1)$ or $x = (0, -1, -1)$,

$$P(x) = (1 - \alpha)\frac{\alpha^2}{4}(1 + a^2), \quad P'(x) = (1 - \alpha)\frac{\alpha^2}{4}\left(1 + \left(\frac{a}{1 - 2\epsilon}\right)^2\right).$$

(c) If $x = (0, 1, -1)$ or $x = (0, -1, 1)$,

$$P(x) = (1 - \alpha)\frac{\alpha^2}{4}(1 - a^2), \quad P'(x) = (1 - \alpha)\frac{\alpha^2}{4}\left(1 - \left(\frac{a}{1 - 2\epsilon}\right)^2\right).$$

(d) If $x = (1, 1, 1)$ or $x = (-1, -1, -1)$,

$$P(x) = \frac{\alpha^3}{8}(1 + 2a + a^2), \quad P'(x) = \frac{\alpha^3}{8}\left(1 + 2a + \left(\frac{a}{1 - 2\epsilon}\right)^2\right).$$

(e) If $x = (1, -1, -1)$ or $x = (-1, 1, 1)$,

$$P(x) = \frac{\alpha^3}{8}(1 - 2a + a^2), \quad P'(x) = \frac{\alpha^3}{8}\left(1 - 2a + \left(\frac{a}{1 - 2\epsilon}\right)^2\right).$$

(f) Otherwise,

$$P(x) = \frac{\alpha^3}{8}(1 - a^2), \quad P'(x) = \frac{\alpha^3}{8}\left(1 - \left(\frac{a}{1 - 2\epsilon}\right)^2\right).$$

Observing that

$$\left(\frac{a}{1 - 2\epsilon}\right)^2 - a^2 = 4\epsilon(1 - \epsilon)\left(\frac{a}{1 - 2\epsilon}\right)^2 \leq 4\epsilon\left(\frac{a}{1 - 2\epsilon}\right)^2 \leq 16a^2\epsilon,$$

we get in cases (b)-(c),

$$\frac{(P(x) - P'(x))^2}{P(x)} \leq \frac{64\alpha^2 a^4 \epsilon^2}{1 - a},$$

and in cases (d)-(f),

$$\frac{(P(x) - P'(x))^2}{P(x)} \leq \frac{32\alpha^2 a^4 \epsilon^2}{1 - a}.$$

Summing, we obtain

$$D(P'||P) \leq \frac{512\alpha^2 a^4 \epsilon^2}{1-a}.$$

Now let $t \leq T_1$ with $c_1 = \frac{1}{512}$. Denoting by $P^t$, $P'^t$ the distributions of $X(1), \ldots, X(t)$ under the respective parameters $\theta, \theta'$, we obtain $D(P'^t||P^t) = tD(P'||P) \leq \ln\left(\frac{1}{4\delta}\right)$. Since $||\theta - \theta'||_\infty = 2\epsilon$, it follows from [Tsybakov, 2008, Theorem 2.2] that:

$$\min\left(\mathbb{P}_\theta(||\hat\theta - \theta||_\infty \geq \epsilon), \mathbb{P}_{\theta'}(||\hat\theta - \theta'||_\infty \geq \epsilon)\right) \geq \frac{1}{4\exp(D(P'^t||P^t))} \geq \delta.$$

Since $\theta, \theta' \in \Theta_{a,b}$, we get

$$\min_{\theta \in \Theta a, b} \mathbb{P}\left(||\hat\theta - \theta||_\infty \geq \epsilon\right) \geq \delta.$$

Now assume that $n > 4$ so that $T_2 > 0$. Let $m = n - 4$ and $c = b/m$. Consider the two parameters

$$\theta = (a, a, -a, -a, c, \ldots, c), \quad \theta' = (-a, -a, a, a, c, \ldots, c).$$

Observe that $\theta, \theta' \in \Theta_{a,b}$. Denote by $P$ and $P'$ the distributions of $X$ under parameters $\theta, \theta'$, respectively. Again, we use Lemma 1 to get an upper bound on the Kullback-Leibler divergence between $P$ and $P'$.

Define $y = (1, 1, -1, -1)$, $k^+ = \sum_{i=1}^4 \mathbf{1}\{x_i = y_i\}$, $k^- = \sum_{i=1}^4 \mathbf{1}\{x_i = -y_i\}$, $k = k^+ + k^-$, $h = k^+ - k^- = x_1 + x_2 - x_3 - x_4$, $\ell^+ = \sum_{i>4} 1_{\{x_i=+1\}}$ and $\ell^- = \sum_{i>4} 1_{\{x_i=-1\}}$, and $\ell = \ell^+ + \ell^-$. We have:

$$P(x) = \frac{1}{2^{k+\ell}}\alpha^{k+\ell}(1-\alpha)^{n-k-\ell}\left((1+a)^{k^+}(1-a)^{k^-}p_{\ell^+,\ell^-} + (1+a)^{k^-}(1-a)^{k^+}p_{\ell^-,\ell^+}\right),$$

$$= \frac{1}{2^{k+\ell}}\alpha^{k+\ell}(1-\alpha)^{n-k-\ell}(1+a)^{k^-}(1-a)^{k^-}\left((1+a)^h p_{\ell^+,\ell^-} + (1-a)^h p_{\ell^-,\ell^+}\right).$$

and

$$P'(x) = \frac{1}{2^{k+\ell}}\alpha^{k+\ell}(1-\alpha)^{n-k-\ell}\left((1+a)^{k^+}(1-a)^{k^-}p_{\ell^-,\ell^+} + (1+a)^{k^-}(1-a)^{k^+}p_{\ell^+,\ell^-}\right),$$

$$= \frac{1}{2^{k+\ell}}\alpha^{k+\ell}(1-\alpha)^{n-k-\ell}(1+a)^{k^-}(1-a)^{k^-}\left((1+a)^h p_{\ell^-,\ell^+} + (1-a)^h p_{\ell^+,\ell^-}\right),$$

where

$$\forall i, j \in \mathbb{N}, \quad p_{i,j} = (1+c)^i(1-c)^j,$$

Define:

$$F(x) = \frac{(P(x) - P'(x))^2}{P(x)P'(x)}$$

We have:

$$F(x) = \frac{((1+a)^h - (1-a)^h)^2(p_{\ell^-,\ell^+} - p_{\ell^+,\ell^-})^2}{((1+a)^h p_{\ell^-,\ell^+} + (1-a)^h p_{\ell^+,\ell^-})((1+a)^h p_{\ell^+,\ell^-} + (1-a)^h p_{\ell^-,\ell^+})}$$

Notice that $F$ is invariant (a) when $h$ is replaced by $-h$ and (b) when one exchanges $\ell^+$ and $\ell^-$. So we can assume that $h > 0$ and $\ell^+ \geq \ell^-$, so that $p_{\ell^+,\ell^-} \geq p_{\ell^-,\ell^+}$. Now:

$$F(x) \leq \frac{((1+a)^h - (1-a)^h)^2}{(1-a)^h(1+a)^h}\frac{(p_{\ell^+,\ell^-} - p_{\ell^-,\ell^+})^2}{p_{\ell^+,\ell^-}^2}.$$

Define $\eta = (1-c)/(1+c)$. Then:

$$\frac{(p_{\ell^+,\ell^-} - p_{\ell^-,\ell^+})^2}{p_{\ell^+,\ell^-}^2} = \left(1 - \eta^{\ell^+-\ell^-}\right)^2 \leq (\ell^+-\ell^-)^2(1-\eta)^2 = (\ell^+-\ell^-)^2\frac{4c^2}{(1+c)^2} \leq 4c^2(\ell^+-\ell^-)^2.$$

Moreover, using the fact that $h \leq 4$,

$$\frac{((1+a)^h - (1-a)^h)^2}{(1-a)^h(1+a)^h} \leq \frac{((1+a)^4 - (1-a)^4)^2}{(1-a)^4} = \frac{(4a(1+a^2))^2}{(1-a)^4} \leq 64\frac{a^2}{(1-a)^4}.$$

By Lemma 1,

$$D(P||P') \leq \mathbb{E}(F(X)) \leq 256\frac{a^2c^2}{(1-a)^4}\mathbb{E}(h^2(X)(\ell^+(X) - \ell^-(X))^2),$$

where $X$ has distribution $P$ and we make explicit the dependency of $h, \ell^+, \ell^-$ in the state $X$. The random quantity $h^2(X)(\ell^+(X) - \ell^-(X))^2$ does not change when $X$ is replaced by $-X$, so:

$$\mathbb{E}(h^2(X)(\ell^+(X) - \ell^-(X))^2) = \mathbb{E}(h^2(X))\mathbb{E}((\ell^+(X) - \ell^-(X))^2).$$

Now

$$\mathbb{E}(h^2(X)) = \mathrm{var}(h(X)) = \mathrm{var}(h(X)|G = 1) = 4\mathrm{var}(X_1|G = 1) = 4\alpha(1 - \alpha a^2)$$

and

$$\mathbb{E}((\ell^+(X) - \ell^-(X))^2) = \mathrm{var}(\ell^+(X) - \ell^-(X)),$$
$$= \mathrm{var}(\ell^+(X) - \ell^-(X)|G = 1),$$
$$= m\mathrm{var}(X_5|G = 1) = m\alpha(1 - \alpha c^2)$$

so that

$$D(P||P') \leq 1024\frac{m\alpha^2 a^2 c^2}{(1-a)^4} = 1024\frac{\alpha^2 a^2 b^2}{m(1-a)^4}.$$

Now let $t \leq T_2$ with $c_2 = \frac{1}{1024}$. Denoting by $P^t, P'^t$ the distributions of $X(1), \ldots, X(t)$ under the respective parameters $\theta, \theta'$, we obtain $D(P^t||P'^t) = tD(P||P') \leq \ln\left(\frac{1}{4\delta}\right)$. Since $||\theta - \theta'||_\infty = 2a$, it follows from [Tsybakov, 2008, Theorem 2.2] that:

$$\min\left(\mathbb{P}_\theta(||\hat{\theta} - \theta||_\infty \geq a), \mathbb{P}_{\theta'}(||\hat{\theta} - \theta'||_\infty \geq a)\right) \geq \frac{1}{4\exp(D(P^t||P'^t))} \geq \delta.$$

Since $\theta, \theta' \in \Theta_{a,b}$ and $a \geq \epsilon$, we get

$$\min_{\theta \in \Theta_{a,b}} \mathbb{P}\left(||\hat{\theta} - \theta||_\infty \geq \epsilon\right) \geq \delta.$$

$\square$

## 2 Proof of Theorem 2

We use the following preliminary results.

**A concentration inequality.** Define:

$$||\hat{C} - C||_\infty = \max_{i,j:i\neq j}|\hat{C}_{ij} - C_{ij}|.$$

**Lemma 2** *We have:*
*(i) For all $i = 1, \ldots, n$ and all $\varepsilon > 0$,*

$$\mathbb{P}(|\sum_{j\neq i}(\hat{C}_{ij} - C_{ij})| \geq \varepsilon) \leq 2\exp\left(-\frac{\varepsilon^2\alpha^2 t}{30\max(B(\theta)^2, n)}\right) + 2n\exp\left(-\frac{t\alpha^2}{8(n-1)}\right).$$

*(ii) For all $j \neq i$ and all $\varepsilon \in (0, 1)$,*

$$\mathbb{P}(|\hat{C}_{ij} - C_{ij}| \geq \varepsilon) \leq 2\exp\left(-\frac{\varepsilon^2\alpha^2 t}{120}\right) + 4\exp\left(-\frac{t\alpha^2}{8}\right).$$

*(iii) For all $\varepsilon \in (0, 1)$,*

$$\mathbb{P}(||\hat{C} - C||_\infty \geq \varepsilon) \leq 3n^2\exp\left(-\frac{\varepsilon^2\alpha^2 t}{120}\right).$$

*Proof.* We first prove (i). Let $\bar{Z} = \sum_{j \neq i}(\hat{C}_{ij} - C_{ij})$, for some fixed $i$. The distribution of $\bar{Z}$ is independent of $G(1), ..., G(t)$ so we fix $G(1) = ... = G(t) = 1$ until the end of the proof. Note that, given $G(t) = 1$, the random variables $X_1(t), ..., X_n(t)$ are independent, with respective expectations $\theta_1, ..., \theta_n$. Let $U = (U_j(t))_{j,t}$ be i.i.d Bernoulli random variables with $\mathbb{E}(U_j(t)) = \alpha$ and $V = (V_j(t))_{j,t}$ be independent random variables on $\{-1, 1\}$ with $\mathbb{E}(V_j(t)) = \theta_j$. Then $(X_j(t))_{j,t}$ has the same distribution as $(U_j(t)V_j(t))_{j,t}$. Define:

$$Z(s) = \frac{U_i(s)U_j(s)V_i(s)V_j(s)}{N_j},$$

with

$$N_j = \sum_{s=1}^{t} U_i(s)U_j(s).$$

Observe that $\bar{Z}$ has the same distribution as $\sum_{s=1}^{t} Z(s)$.

Conditioning

Let us first condition on $U$. We denote by $\mathbb{E}_U$ and $\mathbb{P}_U$ the corresponding conditional expectation and probability. We upper bound the cumulant generating function of $Z(s)$. Consider $s$ fixed and drop $s$ for clarity. Consider $\lambda \in \mathbb{R}$ and:

$$
\begin{aligned}
\ln(\mathbb{E}_U(e^{\lambda Z}|V_i)) &= \ln(\mathbb{E}_U(e^{\lambda U_i V_i \sum_{j \neq i} U_j V_j / N_j}|V_i)) \\
&= \sum_{j \neq i} \ln(\mathbb{E}_U(e^{\lambda U_i U_j V_i V_j / N_j}|V_i)) \\
&\leq \sum_{j \neq i} \ln(\mathbb{E}_U(e^{\lambda U_i U_j V_i \theta_j / N_j + \lambda^2 U_i U_j/(2N_j^2)}|V_i)) \\
&= \lambda^2 \sum_{j \neq i} \frac{U_i U_j}{2N_j^2} + \ln(\mathbb{E}_U(e^{\lambda U_i V_i \sum_{j \neq i} U_j \theta_j / N_j}|V_i)),
\end{aligned}
$$

using the independence of $V_1, ..., V_n$, and the fact that, if $Y$ is a random variable with $|Y| \leq 1$, then $\ln(\mathbb{E}(e^{\lambda Y})) \leq \lambda\mathbb{E}(Y) + \lambda^2/2$ by Hoeffding's lemma. Taking expectation over $V_i$:

$$
\ln(\mathbb{E}_U(e^{\lambda U_i V_i \sum_{j \neq i} U_j \theta_j / N_j})) \leq \lambda \sum_{j \neq i} \frac{U_i U_j \theta_i \theta_j}{N_j} + \frac{\lambda^2}{2}\left(\sum_{j \neq i} \frac{U_i U_j \theta_j}{N_j}\right)^2,
$$

where we have used Hoeffding's lemma once again. Putting it together, we have proven:

$$
\ln(\mathbb{E}_U(e^{\lambda Z})) \leq \lambda \sum_{j \neq i} \frac{U_i U_j \theta_i \theta_j}{N_j} + \frac{\lambda^2}{2}\left(\sum_{j \neq i} \frac{U_i U_j}{N_j^2} + \left(\sum_{j \neq i} \frac{U_i U_j \theta_j}{N_j}\right)^2\right).
$$

Define $N = \min_{j \neq i} N_j$, $S = \sum_{s=1}^{t}\left(\sum_{j \neq i} U_i(s)U_j(s)\theta_j\right)^2$ and $\sigma^2 = \frac{(n-1)N+S}{N^2}$. It is noted that $N$, $S$ and $\sigma^2$ depend on $(U_j(t))_{j,t}$ but not on $(V_j(t))_{j,t}$.

Using independence,

$$
\begin{aligned}
\ln(\mathbb{E}_U(e^{\lambda \bar{Z}})) &= \sum_{s=1}^{t} \ln(\mathbb{E}_U(e^{\lambda Z(s)})), \\
&\leq \lambda\mathbb{E}(\bar{Z}) + \frac{\lambda^2}{2}\left(\sum_{j \neq i} \frac{1}{N_j} + \sum_{s=1}^{t}\left(\sum_{j \neq i} \frac{U_i(s)U_j(s)\theta_j}{N_j}\right)^2\right), \\
&\leq \lambda\mathbb{E}(\bar{Z}) + \frac{\lambda^2 \sigma^2}{2},
\end{aligned}
$$

where we used the fact that $\mathbb{E}_U(\bar{Z}) = \mathbb{E}(\bar{Z}) = \theta_i \sum_{j \neq i} \theta_j$.

## Chernoff bound

We now derive a Chernoff bound for $\bar{Z}$. For all $\varepsilon > 0$, we have

$$\mathbb{P}_U(\bar{Z} - \mathbb{E}(\bar{Z}) \geq \varepsilon) \leq \min_{\lambda \geq 0} e^{-\lambda \varepsilon} \mathbb{E}_U(e^{\lambda(\bar{Z} - \mathbb{E}(\bar{Z}))}) \leq \min_{\lambda \geq 0} e^{-\lambda \varepsilon + \lambda^2 \sigma^2/2} = e^{-\varepsilon^2/\sigma^2},$$

the minimum being attained for $\lambda = 2\varepsilon/\sigma^2$.

## Controlling the fluctuations of $\sigma^2$

To remove the conditioning on $U$, so that we need to control the fluctuations of $\sigma^2$, thus those of $N$ and $S$. First consider $N$. Since $N_j$ is the sum of $t$ independent Bernoulli variables with expectation $\alpha^2$, we have by Lemma 3,

$$\mathbb{P}(N_j \leq \alpha^2 t/2) \leq e^{-\frac{t\alpha^2}{8}}.$$

Using a union bound,

$$\mathbb{P}(N \leq \alpha^2 t/2) \leq \sum_{j \neq i} \mathbb{P}(N_j \leq \alpha^2 t/2) \leq (n-1) e^{-\frac{t\alpha^2}{8}}.$$

We turn to $S$. $S$ is a sum of $t$ positive independent variables bounded by $(n-1)^2$ with expectation

$$\mu = \mathbb{E}\left(\sum_{j \neq i} U_i(t) U_j(t) \theta_j\right)^2 = \alpha^2(\alpha B_i(\theta)^2 + (1-\alpha) \sum_{j \neq i} \theta_j^2),$$

where $B_i(\theta) = \sum_{j \neq i} \theta_j$. We have $\mu \leq \bar{\mu} \equiv \alpha^2 \max(B_i(\theta)^2, n-1)$. By Lemma 3,

$$\begin{aligned}
\mathbb{P}(S \geq 2t\bar{\mu}) = \mathbb{P}\left(\frac{S}{(n-1)^2} \geq \frac{2t\bar{\mu}}{(n-1)^2}\right) \\
\leq \exp\left(-tD\left(\frac{2\bar{\mu}}{(n-1)^2} \Big\| \frac{\mu}{(n-1)^2}\right)\right) \\
\leq \exp\left(-tD\left(\frac{2\bar{\mu}}{(n-1)^2} \Big\| \frac{\bar{\mu}}{(n-1)^2}\right)\right) \\
\leq e^{-\frac{t\bar{\mu}}{3(n-1)^2}} \\
\leq e^{-\frac{t\alpha^2}{3(n-1)}}.
\end{aligned}$$

If both events $S \leq 2t\bar{\mu}$ and $N \geq \alpha^2 t/2$ occur we have:

$$\sigma^2 \leq \frac{2(n-1)}{\alpha^2 t} + \frac{8 \max(B_i(\theta)^2, n-1)}{\alpha^2 t} \leq \frac{10 \max(B_i(\theta)^2, n-1)}{\alpha^2 t}.$$

## Estimation Error

Finally,

$$\begin{aligned}
\mathbb{P}(\bar{Z} - \mathbb{E}(\bar{Z}) \geq \varepsilon) \leq \mathbb{P}\left(\bar{Z} - \mathbb{E}(\bar{Z}) \geq \varepsilon, \sigma^2 \leq \frac{10 \max(B_i(\theta)^2, n-1)}{\alpha^2 t}\right) + \mathbb{P}(N \leq \alpha^2 t/2) + \mathbb{P}(S \geq 2t\bar{\mu}) \\
\leq \exp\left(-\frac{\varepsilon^2 \alpha^2 t}{10 \max(B_i(\theta)^2, n-1)}\right) + (n-1)\exp\left(-\frac{t\alpha^2}{8}\right) + \exp\left(-\frac{t\alpha^2}{3(n-1)}\right) \\
\leq \exp\left(-\frac{\varepsilon^2 \alpha^2 t}{10 \max(B_i(\theta)^2, n-1)}\right) + n \exp\left(-\frac{t\alpha^2}{8(n-1)}\right).
\end{aligned}$$

Doing the same reasoning for $\mathbb{P}(\bar{Z} - \mathbb{E}(\bar{Z}) \leq -\varepsilon)$ yields

$$\mathbb{P}(|\bar{Z} - \mathbb{E}(\bar{Z})| \geq \varepsilon) \leq 2\exp\left(-\frac{\varepsilon^2 \alpha^2 t}{10 \max(B_i(\theta)^2, n-1)}\right) + 2n \exp\left(-\frac{t\alpha^2}{8(n-1)}\right).$$

Statement (i) then follows from the fact that $\max(B_i(\theta)^2, n-1) \leq 3 \max(B(\theta)^2, n)$. Indeed, $\max(B_i(\theta)^2, n-1) \leq \max(B_i(\theta)^2, 3n)$, and if $B_i(\theta) \geq \sqrt{3n} \geq 3$,

$$\frac{B_i(\theta)^2}{B(\theta)^2} \leq \frac{B_i(\theta)^2}{(B_i(\theta) - 1)^2} \leq \frac{9}{4} \leq 3.$$

Statement (ii) is obtained by setting $n = 2$ in statement (i); statement (iii) follows from a union bound over all pairs $i, j$ of statement (ii), on observing that

$$\mathbb{P}(|\hat{C}_{ij} - C_{ij}| \geq \varepsilon) \leq 6 \exp\left(-\frac{\varepsilon^2 \alpha^2 t}{120}\right).$$

$\square$

**Lemma 3 (Chernoff's Inequality)** *Let $Y_1, ..., Y_t$ be i.i.d. random variables on $[0, 1]$ with expectation $\mu$. Denote by $D(\mu'||\mu)$ the Kullback Leibler divergence between two Bernoulli distribution with parameters $\mu'$ and $\mu$.*

*(i) For all $\mu' \geq \mu$, $\mathbb{P}(\sum_{s=1}^{t} Y_s \geq t\mu') \leq e^{-tD(\mu'||\mu)}$.*

*(ii) For all $\mu' \leq \mu$, $\mathbb{P}(\sum_{s=1}^{t} Y_s \leq t\mu') \leq e^{-tD(\mu'||\mu)}$.*

*(iii) For all $\mu \geq 0$, $D(2\mu||\mu) \geq \mu/2$ and $D(\mu/2||\mu) \geq \mu/8$.*

**Estimation of absolute value.** For all $i \neq j \neq k$, define

$$\rho_k(i, j) = \sqrt{\left|\frac{\hat{C}_{ik}\hat{C}_{jk}}{\hat{C}_{ij}}\right|}.$$

**Lemma 4** *If $||\hat{C} - C||_\infty \leq \varepsilon$, then*

$$|\rho_k(i, j) - |\theta_k|| \leq 10\frac{\varepsilon}{|C_{ij}|}.$$

*Proof.* Without loss of generality, assume that $|\theta_i| \geq |\theta_j|$ so that $|C_{ik}| \geq |C_{jk}|$.

a) If $\varepsilon \geq |C_{ij}|/2$, the inequality holds since $10\frac{\varepsilon}{|C_{ij}|} \geq 5$ and $|\rho_k(i, j) - |\theta_k|| \leq 2$.

b) Assume $\varepsilon \geq C_{ij}/2$ and $\varepsilon \geq C_{jk}/2$. Then

$$\theta_k = C_{jk}/|\theta_j| \leq 2\varepsilon/|\theta_j| \leq 2\varepsilon/|C_{ij}|.$$

Furthermore, $|\hat{C}_{jk}| \leq |C_{jk}| + \varepsilon$, $|\hat{C}_{ik}| \leq |C_{jk}| + \varepsilon$ and $|\hat{C}_{ij}| \geq |C_{ij}| - \varepsilon \geq |C_{ij}|/2$. So

$$\rho_k(i, j) \leq \sqrt{2(|C_{ik}| + \varepsilon)(|C_{jk}| + \varepsilon)/C_{ij}} = \sqrt{2(|\theta_k| + \frac{\varepsilon}{|\theta_i|})(|\theta_k| + \frac{\varepsilon}{|\theta_j|})}$$

$$\leq 2|\theta_k| + 2\varepsilon(\frac{1}{|\theta_i|} + \frac{1}{|\theta_i|}) \leq 2|\theta_k| + \frac{4\varepsilon}{|C_{ij}|}$$

and

$$|\rho_k(i, j) - |\theta_k|| \leq |\rho_k(i, j)| + |\theta_k| \leq 3|\theta_k| + \frac{4\varepsilon}{|C_{ij}|} \leq 10\frac{\varepsilon}{|C_{ij}|}.$$

c) Finally, let $\varepsilon \leq \min(|C_{ij}|, |C_{ik}|, |C_{jk}|)/2$. Define

$$\Delta = |C_{ik}C_{jk}\hat{C}_{ij} - \hat{C}_{ik}\hat{C}_{jk}C_{ij}|.$$

We have

$$\Delta \leq |C_{ik}C_{jk}||\hat{C}_{ij} - C_{ij}| + |C_{ij}\hat{C}_{ik}||\hat{C}_{jk} - C_{jk}| + |C_{ij}C_{jk}||\hat{C}_{ik} - C_{ik}|,$$
$$\leq \varepsilon(|C_{ik}C_{jk}| + 2|C_{ij}C_{ik}| + |C_{ij}C_{jk}|).$$

Further,

$$|\rho_k(i, j)^2 - \theta_k^2| = \frac{\Delta}{|\hat{C}_{ij}C_{ij}|} \leq \frac{2\Delta}{C_{ij}^2} \leq \frac{2\varepsilon}{|C_{ij}|}(\theta_k^2 + 2|\theta_i\theta_k| + |\theta_j\theta_k|) \leq \frac{8\varepsilon|\theta_k|}{|C_{ij}|}.$$

Finally, $\rho_k(i, j)^2 - \theta_k^2 = (\rho_k(i, j) + |\theta_k|)(\rho_k(i, j) - |\theta_k|)$, so that

$$||\rho_k(i, j)| - |\theta_k|| \leq \frac{|\rho_k(i, j)^2 - \theta_k^2|}{\rho_k(i, j) + |\theta_k|} \leq \frac{|\rho_k(i, j)^2 - \theta_k^2|}{|\theta_k|} \leq \frac{8\varepsilon}{|C_{ij}|}.$$

$\square$

**Lemma 5** *If* $||\hat{C} - C||_\infty \le \varepsilon$, *then*

$$||\hat{\theta}_k| - |\theta_k|| \le \frac{20\varepsilon}{A^2(\theta)}.$$

*Proof.* a) If $\varepsilon \ge A^2(\theta)/4$,

$$||\hat{\theta}_k| - |\theta_k|| \le 2 \le 5 \le \frac{20\varepsilon}{A^2(\theta)}$$

so that the inequality holds.

b) Consider $\varepsilon \le A^2(\theta)/4$. By definition, $\hat{\theta}_k = \rho_k(i_k, j_k)$. By assumption, there exists $i, j \ne k$ such that $C_{i,j} \ge A^2(\theta)$. Further:

$$|C_{i_k,j_k}| + \varepsilon \ge |\hat{C}_{i_k,j_k}| \ge |\hat{C}_{i,j}| \ge |C_{i,j}| - \varepsilon \ge A^2(\theta) - \varepsilon.$$

So $C_{i_k,j_k} \ge A^2(\theta) - 2\varepsilon \ge A^2(\theta)/2$. Using Lemma 4,

$$||\hat{\theta}_k| - |\theta_k|| = |\rho_k(i_k, j_k) - |\theta_k|| \le \frac{10\varepsilon}{C_{i_k j_k}} \le \frac{20\varepsilon}{A^2(\theta)}.$$

$\square$

**Sign estimation.** We will use the following fact:

**Fact 1** *Let* $u, v \in \mathbb{R}^n$ *and define* $\epsilon = \max_i |u_i - v_i|$, $\bar{u} = \max_i |u_i|$, $i^* \in \arg\max_i |v_i|$. *If* $\epsilon \le \bar{u}/4$ *then* (i) $\operatorname{sign}(v_{i^*}) = \operatorname{sign}(u_{i^*})$ *and* (ii) $|u_{i^*}| \ge \bar{u}/2$.

*Proof.* (i) We proceed by contradiction. Assume that $\operatorname{sign}(u_{i^*}) \ne \operatorname{sign}(v_{i^*})$. Since $i^* \in \arg\max_i |v_i|$, we have $|v_{i^*}| \ge \bar{u} - \epsilon$. On the other hand, since $\operatorname{sign}(u_{i^*}) \ne \operatorname{sign}(v_{i^*})$, $|v_j| \le \epsilon$. Hence $2\epsilon \ge \bar{u}$, a contradiction.

(ii) We have $|u_{i^*}| \ge v_{i^*} - \epsilon \ge \max_i |v_i| - \epsilon \ge \bar{u} - 2\epsilon \ge \bar{u}/2$. $\square$

In the rest of the proof, we define $\phi = \max_i |\theta_i|$.

**Lemma 6** *Assume that* $||\hat{C} - C||_\infty \le \varepsilon \le \frac{A^2(\theta)}{2}$. *Then for all* $k = 1, \ldots, n$,

$$|\hat{\theta}_k^2 - \theta_k^2| \le \frac{8\varepsilon\phi^2}{A^2(\theta)}.$$

*Proof.* We have

$$|\rho_k(i,j)^2 - \theta_k^2| = \frac{\Delta}{|\hat{C}_{ij} C_{ij}|},$$

with

$$\Delta = |C_{ik} C_{jk} \hat{C}_{ij} - \hat{C}_{ik} \hat{C}_{jk} C_{ij}|.$$

Now,

$$\Delta \le |C_{ik} C_{jk}||\hat{C}_{ij} - C_{ij}| + |C_{ij}\hat{C}_{ik}||\hat{C}_{jk} - C_{jk}| + |C_{ij} C_{jk}||\hat{C}_{ik} - C_{ik}|,$$

$$\le \varepsilon(|C_{ik} C_{jk}| + |C_{ij}\hat{C}_{ik}| + |C_{ij} C_{jk}|),$$

since $||\hat{C} - C||_\infty \le \varepsilon$. We have $|\hat{C}_{ik}| \le |C_{ik} + \varepsilon| \le \phi^2 + \varepsilon \le 2\phi^2$, since $\phi^2 \ge A^2(\theta) \ge \varepsilon$. Further, using the fact that $|C_{ik} C_{jk}| \le \phi^2 |C_{ij}|$, we get $|C_{ij} C_{jk}| \le \phi^2 |C_{ij}|$. Replacing, we obtain

$$\Delta \le 4\phi^2 \varepsilon |C_{ij}|$$

and

$$|\rho_k(i,j)^2 - \theta_k^2| = \frac{4\phi^2\varepsilon}{|\hat{C}_{ij}|}.$$

We have

$$\max_{i,j \ne k} |\hat{C}_{ij}| \ge \max_{i,j \ne k} |C_{ij}| - \varepsilon \ge A^2(\theta) - A^2(\theta)/2 = A^2(\theta)/2.$$

Setting $(i,j) \in \arg\max_{i,j \ne k} |\hat{C}_{ij}|$, we get the announced result. $\square$

**Estimation error.** We can now control the estimation error.

**Lemma 7** *Assume that* $||\hat{C} - C||_\infty \le \varepsilon \le A^2(\theta)\min(\frac{1}{2}, \frac{B(\theta)}{64})$ *and* $\max_i |\sum_{j\ne i} \hat{C}_{ij} - C_{ij}| \le \frac{A(\theta)B(\theta)}{8}$. *Then*

*(i)* $\text{sign}(\hat{\theta}_{k^*}) = \text{sign}(\theta_{k^*})$,

*(ii) for all $i$ such that $|\theta_i| \ge \frac{2\varepsilon}{A(\theta)}$,* $\text{sign}(\hat{\theta}_i) = \text{sign}(\theta_i)$,

*(iii)* $||\hat{\theta} - \theta||_\infty \le \frac{24\varepsilon}{A^2(\theta)}$.

*Proof.* (i) Let $u_i = \theta_i^2 + \sum_{j\ne i} C_{ij} = \theta_i B(\theta)$ and $v_i = \hat{\theta}_i^2 + \sum_{j\ne i} \hat{C}_{ij}$. We have

$$|u_i - v_i| \le |\hat{\theta}_i^2 - \theta_i^2| + |\sum_{j\ne i}(\hat{C}_{ij} - C_{ij})|.$$

Using Lemma 6,

$$|\hat{\theta}_i^2 - \theta_i^2| \le \min\left(\frac{\phi^2 B(\theta)}{8}, 4\phi^2\right) \le \frac{\phi^2 B(\theta)}{8} \le \frac{\phi B(\theta)}{8}.$$

Further, since $A(\theta) \le \phi$,

$$|\sum_{j\ne i}(\hat{C}_{ij} - C_{ij})| \le \frac{\phi B(\theta)}{8}.$$

So, for all $i$,

$$|u_i - v_i| \le \frac{\phi B(\theta)}{4} = \frac{\max_i |u_i|}{4}.$$

Applying Fact 1 statement (i) ensures that $\text{sign}(\hat{\theta}_{k^*}) = \text{sign}(\theta_{k^*})$.

(ii) Fact 1 statement (ii) gives $|\theta_{k^*}|B(\theta) \ge \frac{\phi B(\theta)}{2}$, so that $|\theta_{k^*}| \ge \phi/2 \ge A(\theta)/2$. Consider $i \ne k^*$ and $|\theta_i| \ge 2\varepsilon/A(\theta)$. We have $|C_{ik^*}| = |\theta_i||\theta_{k^*}| \ge \varepsilon$ since $|\theta_{k^*}| \ge A(\theta)/2$. Since $|\hat{C}_{ik^*} - C_{ik^*}| \le \varepsilon$, we have $\text{sign}(\hat{C}_{ik^*}) = \text{sign}(C_{ik^*})$. So $\text{sign}(\hat{\theta}_i) = \text{sign}(\theta_i)$ which proves the second claim.

(iii) We have:

$$|\hat{\theta}_i - \theta_i| \le ||\hat{\theta}_i| - |\theta_i|| + 2|\theta_i|\mathbf{1}\{\text{sign}(\theta_i) \ne \text{sign}(\hat{\theta}_i)\} \le \frac{20\varepsilon}{A^2(\theta)} + \frac{4\varepsilon}{A(\theta)} \le \frac{24\varepsilon}{A^2(\theta)}.$$

where we applied the previous statement and Lemma 5. $\qquad\square$

**Proof of Theorem 2.** Let $\theta \in \Theta_{a,b}$, $\epsilon \in (0, \min(b/3, 1))$ and assume that the following two events occur:

$$\left\{||\hat{C} - C||_\infty \le \frac{\epsilon A^2(\theta)}{24}\right\} \text{ and } \left\{\max_{i=1,\dots,n}\left|\sum_{j\ne i}(\hat{C}_{ij} - C_{ij})\right| \le \frac{A(\theta)B(\theta)}{8}\right\}.$$

We may readily check that

$$\frac{\epsilon A^2(\theta)}{24} \le \frac{A^2(\theta)}{24}\min\left(\frac{b}{3}, 1\right) \le \frac{A^2(\theta)}{24}\min\left(\frac{B(\theta)}{3}, 1\right) \le A^2(\theta)\min\left(\frac{B(\theta)}{64}, \frac{1}{2}\right).$$

Thus Lemma 7 guarantees that $||\hat{\theta} - \theta||_\infty \le \epsilon$. By a union bound,

$$\mathbb{P}(||\hat{\theta} - \theta||_\infty \ge \epsilon) \le \mathbb{P}\left(||\hat{C} - C||_\infty \ge \frac{\epsilon A^2(\theta)}{24}\right) + \sum_{i=1}^n \mathbb{P}\left(\left|\sum_{j\ne i}(\hat{C}_{ij} - C_{ij})\right| \ge \frac{A(\theta)B(\theta)}{8}\right).$$

Now let $t \ge \max(T_1', T_2')$ with $c_1' = 120 \times 24^2$ and $c_2' = 30 \times 8^2$. Applying Lemma 2,

$$\mathbb{P}\left(||\hat{C} - C||_\infty \ge \frac{\epsilon A^2(\theta)}{24}\right) \le 3n^2 \exp\left(-\frac{\epsilon^2 A^4(\theta)\alpha^2 t}{120 \times 24^2}\right) \le \frac{\delta}{2}.$$

Applying Lemma 2 once again, we get

$$\mathbb{P}\left(\left|\sum_{j\neq i}(\hat{C}_{ij}-C_{ij})\right|\geq\frac{A(\theta)B(\theta)}{8}\right)\leq 2\exp\left(-\frac{A(\theta)^2B(\theta)^2\alpha^2 t}{30\times 8^2\max\left(B(\theta)^2,n\right)}\right)+2n\exp\left(-\frac{t\alpha^2}{8(n-1)}\right).$$

Since

$$\frac{A(\theta)^2B(\theta)^2\alpha^2 t}{30\times 8^2\max(B(\theta)^2,n)}\geq\frac{a^2\alpha^2 t}{30\times 8^2}\max\left(1,\frac{b^2}{n}\right)\geq\ln\left(\frac{4n^2}{\delta}\right)$$

and

$$\frac{t\alpha^2}{8(n-1)}\geq\ln\left(\frac{4n^2}{\delta}\right),$$

we get

$$\mathbb{P}\left(\left|\sum_{j\neq i}(\hat{C}_{ij}-C_{ij})\right|\geq\frac{A(\theta)B(\theta)}{8}\right)\leq\frac{\delta}{n^2},$$

and summing we get $\mathbb{P}(||\hat{\theta}-\theta||_\infty\geq\epsilon)\leq\delta$ as announced. $\qquad\square$

## 3    Upper bounds: relation with prior work

Consider $n\geq 5$, $a=1/2$ , $b=\frac{\sqrt{n-4}}{2}$ and $\theta=(a,-a,a,-a,\frac{b}{n-4},...,\frac{b}{n-4})$. We have $\max_i|\theta_i|\leq\frac{1}{2}$.

For Theorem 4 of Zhang et al. [2014] to hold, one requires the following inequality to be satisfied:

$$n\geq c_4\frac{\ln(tn/\delta)}{\overline{D}}$$

where $c_4>0$ is a universal constant and:

$$\overline{D}=\frac{1}{n}\sum_{i=1}^n\mathrm{KL}\left(\frac{1+\theta_i}{2},\frac{1-\theta_i}{2}\right),$$

where $\mathrm{KL}(p,q)$ denotes the Kullback-Leibler divergence between Bernoulli distributions with parameters $p$ and $q$. From inequality $\ln(z)\leq z-1$, we have $\mathrm{KL}(p,q)\leq\frac{(p-q)^2}{q(1-q)}$ for all $p,q$ in $(0,1)$. Since $\max_i|\theta_i|\leq 1/2$ we get:

$$\overline{D}\leq\frac{16}{3n}\sum_{i=1}^n\theta_i^2=\frac{20}{3n}.$$

Therefore $n$ must satisfy $n\geq c_4(3n/20)\ln(tn/\delta)$ and there can exist no such $n$ when $t$ or $1/\delta$ are sufficiently large.