[Reviews · NeurIPS 2017]

Reviewer 1



The authors propose a new lower bound and an efficient method to evaluate the reliability of worker in crowdsourcing scenario. The proposed bound is stricter than the previous ones. The method can also be used in the streaming setting and do not need any EM step to reach the optimal bound as previous work did. This paper proposes sound theoretical proof. It clearly points out the advantage comparing to the previous work. The paper conducted experiments on real and simulated datasets. The results do not show overwhelming out-performance (only better in some settings), which indicates certain room for improvement. Nevertheless, it is s solid paper that suggests an interesting research direction.

Reviewer 2



Summary of paper: The authors derive a lower bound on estimating crowdsource worker reliability in labeling data. They then propose an algorithm to estimate this reliability and analyze the algorithm’s performance. Summary of review: This submission presents a solid piece of research that is well-grounded in both theory and application. Detailed review: — Quality: This paper is thorough. The lower bound and relevant proofs are outlined cleanly. The algorithm is presented in conjunction with time and memory complexity analysis, and the authors compare to related work nicely. Both simulated and real data sets are used; the simulated data have multiple settings and six sources of real-world data are used. The experiments do not demonstrate that the proposed algorithms outperforms all competing methods, but these results are refreshingly realistic and provide a compelling argument for the use of the proposed algorithm. — Clarity: The paper is well-written; the problem and contributions are outlined clearly. The paper would benefit from more description regarding the intuitions of TE and perhaps a high-level summary paragraph in the numerical experiments section. — Originality: The proposed algorithm is novel and the derived lower bound is tied nicely to the problem, bringing a rare balance of theory and application. — Significance: While the problem addressed is very specific, it is important. The proposed algorithm does well—even if it doesn’t beat all others in every instance, it’s more consistent in its performance. This paper will be a solid contribution to the crowdsourcing literature, and could easily impact the use of crowdsourced labels as well.

Reviewer 3



The results presented in the paper seem correct. I did not find any flaws in the analysis. The empirical results seem good but not a big improvement over other methods. However the significance of the results presented escapes me, as I am not familiar with the literature or have any practical experience in crowdsourcing.